



# Can controlled drainage control agricultural nutrient emissions? Evidence from a BACI experiment combined with a dual isotope approach

Mette V. Carstensen[1], Jane R. Poulsen[1], Niels B. Ovesen[1], Christen D. Børgesen[2], Søren K. Hvid[3], Brian Kronvang[1]

[1]Department of Bioscience, Aarhus University, Vejlsøvej 25, 8600 Silkeborg, Denmark
[2]Department of Agroecology, Blichers Allé 20, 8830 Tjele, Denmark
[3]SEGES, Agro Food Park 15, 8200 Aarhus N, Denmark

*Correspondence to*: Mette V. Carstensen (mvc@bios.au.dk)

**Abstract.** Controlled drainage combined with winter crops was investigated for the first time as a mitigation measure for reducing nitrate loss through drainage systems during a three-year period (2012-15) on a loamy soil in Denmark. The impact of controlled drainage on groundwater levels, drain flow, total nitrogen, nitrate, ammonium, nitrous oxide, total phosphorous, and phosphate when applying regulation levels of 50 and 70 cm above drain pipes were determined by using a before-after control-impact (BACI) study design. The regulation level had to be 70 cm to significantly elevate groundwater levels and reduce drain outflow and nitrate loss, which were reduced by 38-52 % (97-127 mm) and 36-48 % (6-8 kg nitrate-N ha$^{-1}$) relative to conventional drainage levels, respectively. Controlled drainage did not appear to influence harvest yield or cause pollution swapping as the loss of total phosphorous, phosphate, and nitrous oxide was reduced by 44-45 %, 44-54 %, and 36-38 %, respectively. Stable isotope analysis of $\delta^{15}N$ and $\delta^{18}O$ in nitrate from drain water supported by measurements of nitrate, sulphate, and ammonium concentrations in drain water revealed that denitrification was not enhanced markedly at the impacted plots, except during one event where a pronounced decline in nitrate concentrations and ceased drain flow was observed.

## 1 Introduction

Drainage of fields by subsurface tile drainage pipes is a common farming practice in Denmark, thus, approximately 50 % of the agricultural fields are tile drained, in particular loamy soils (Aslyng, 1980). High amounts of water and different forms of nitrogen (N), especially nitrate ($NO_3^-$-N), are transported directly from the root zone of the fields to nearby streams with the risk of creating eutrophication of freshwater and coastal water bodies, potentially leading to hypoxia events (Rivett et al., 2008). As the diffusive loss of N from agricultural fields is the predominant N source to most Danish coastal waters, mitigation measures aimed to reduce N emissions to surface waters are strongly needed (Jensen et al., 2015). Previous studies have demonstrated that use of controlled drainage on fields with periodical increases of the groundwater table may be a targeted mitigation measure to reduce the loading of $NO_3^-$-N from drain pipes (Wesström and Messing, 2007;Wesström et al., 2001;Lalonde et al., 1996;Drury et al., 2014;Gaynor et al., 2002;Skaggs et al., 2010). Controlled drainage has been tested on a wide range of soil types and under various climate conditions, which is reflected in the large variation – from 21to 96 % (9-30 kg N ha$^{-1}$ yr$^{-1}$) in the efficiency of reducing $NO_3^-$-N loading compared with conventional drainage (Evans et al., 1989;Lalonde et al., 1996;Wesström et al., 2001;Ramoska et al., 2011;Drury et al., 2014;Gilliam et al., 1979;Williams et al., 2015;Fang et al., 2012;Tan et al., 1999;Drury et al., 2009;Sunohara et al., 2014;Gaynor et al., 2002). Most studies



demonstrate that the decreased $NO_3^-$-N loading is primarily a consequences of reduced drain flow, and a few studies document it to be a combination of reduced drain flow and enhanced denitrification (Lalonde et al., 1996;Wesström et al., 2001;Gilliam et al., 1979). Denitrification, which requires anoxic conditions, can potentially be enhanced by controlled drainage as an elevated groundwater table in a field leaves a larger proportion of the soil column saturated, promoting anoxic

conditions. However, if $NO_3^-$-N is not completely reduced to $N_2$, there is a risk of emission of the powerful greenhouse gas, nitrous oxide ($N_2O$-N), to air or water (Knowles, 1982). Consequently, controlled drainage could lead to 'pollution swapping', i.e. substitution of one environmental problem with another. Another concern is related to $PO_4^{3-}$ as reduced conditions can lead to desorption of $PO_4^{3-}$ from hydrous iron oxides if $Fe^{3+}$ is reduced to $Fe^{2+}$ (Hoffmann et al., 2009).

The stable isotope ratios of $\delta^{15}N/\delta^{14}N$ and $\delta^{18}O/\delta^{16}O$ of $NO_3^-$-N have proven to be a powerful tool in detecting denitrification

as microbes prefer light isotopes over heavy isotopes. Field and laboratory studies have documented that denitrification results in a distinct relationship of $\delta^{15}N$ and $\delta^{18}O$ around 0.5-1 in the remaining pool of $NO_3^-$-N (Aravena and Robertson, 1998;Mengis et al., 1999;Li et al., 2014;Fukada et al., 2003;Sigman et al., 2005;Granger et al., 2008).

The aims of this study were to investigate and test the following hypotheses: i) Does controlled drainage assist in reducing $NO_3^-$-N losses to surface waters when cropping a winter wheat?; ii) Does controlled drainage promote pollution swapping by

increasing the indirect emissions of $N_2O$-N and losses of $PO_4^{3-}$ via drainage water?; iii) Can the dual isotope technique using $\delta^{15}N$ and $\delta^{18}O$ of $NO_3^-$-N be used to assess changes in N transformation processes, such as denitrification, when applying controlled drainage?

## 2 Materials and methods

### 2.1 Study area

The experimental site was located in Odder municipality, Denmark (55°55'01.2"N 10°10'54.5"E), and consisted of four adjacent drainage systems draining approximately 1 ha each, with 3-5 lateral drainage pipes placed at app. 15 m intervals and at 1.1 m depth (Fig. 1). A control structure and a measuring well were installed at each plot. The study was conducted during the main drainage period (September to April) of the agro-hydrological years (4/1-3/31) 2012/13 (Y0), 2013/14 (Y1), and 2014/15 (Y2). The first period, Y0, served as reference without application of controlled drainage. In Y1 and Y2

controlled drainage was introduced at two of the plots (IP1 and IP2), while the two other plots served as controls without controlled drainage (CP1 and CP2). In 2013/14, the regulation levels were set to 50 cm above normal drain depth at the regulation well. However, no effect on groundwater levels was detected and the regulation level was therefore increased to 70 cm on 01/28/14 at IP1. In Y2, the regulation level was set to 70 cm at both IP1 and IP2.

The soil samples for texture analysis which were taken adjacent to the regulation wells showed that the plots are

characterized by silty loam and loam (USDA classification) in the root zone (1 m depth) with an average clay, silt and sand content of 14.9±3.7, 21.6±6 and 59.5±7.8 %, respectively (Table S1). The soil C content was 1.7-1.8 % C in the AP horizon (0-27 cm) at all plots (Table S2). For IP2 the C content was low (<0.16 %) in the remaining of the profile, while CP1 and CP2 had a C horizon (95-110 cm) with 2 % C and a high chalk content (7.3-15.5 %). At IP1 an interglacial lake deposit with a high humus and carbon (C) content (6.6 % C) were present at approximately 120-130 cm depth. Field management

practices were similar during the three-year study period involving growth of winter wheat and application of identical amounts of manure and fertilizer in spring.

### 2.2 Water monitoring and modelling

Groundwater levels were monitored continuously at each plot by a pressure transducer, MadgeTech level 2000 (accuracy ±0.3 %), located in the piezometer pipes closest to the regulation well. The piezometer pipes had no filter pack, which

resulted in periodically clogged screens, especially at IP2, thus the continuously measured groundwater levels at IP2 had to





be omitted from the analysis. The groundwater levels were spatially monitored with a water level meter, Solinst 101 (accuracy ±0.3 cm) three to four times per month in eight to nine piezometer pipes with filter packs evenly distributed at each plot. Drain flow was monitored by an electromagnetic flow meter, Waterflux 3000 (accuracy ±0.3%), recording every 10 minutes, and a converter, IFC 100, placed in each measuring well. Ground water storage change (ΔGW) was calculated

by the difference between groundwater level in the beginning and end of the study periods and the drainable porosity of the soil.

Actual evaporation (AE) and percolation to the deeper groundwater was simulated with the Danish root zone model, Daisy (version 5.19), which is a one-dimensional, dynamic model driven by daily weather and farm management data (Abrahamsen and Hansen, 2000). The model has been validated in several international studies and thus holds high

credibility (Refsgaard et al., 1999;Djurhuus et al., 1999;Jensen et al., 1997;Müller et al., 2006;Pedersen et al., 2010). Weather data were obtained from the Danish Meteorological Institute (DMI) as a 10x10 km grid covering the experimental site and corrected according to Allerup and Madsen (1980). Field management data were derived from the fertilizer plans elaborated for each field. Harvest yield was measured by the combined harvester.

### 2.3 Sampling and chemical analysis

Weekly grab samples of drain water were taken in the measuring well, and two to three monthly grab samples of the upper groundwater were taken in the piezometer pipes. Concentrations of nitrate ($NO_3^-$-N), total nitrogen (TN), ammonium ($NH_4^+$-N), and sulphate ($SO_4^{2-}$) were determined by ion chromatography using a Dionex ICS-1500. $NO_3^-$-N and $SO_4^{2-}$, and $NH_4^+$-N were analysed according to ISO 10304/1(Standard, 2009) (detection limit (d.l) 0.050 mgL$^{-1}$) and ISO 11732 (Standard, 2005) (d.l. 0.010 mg L$^{-1}$), respectively. TN analysis followed ISO 12260 (Standard, 2003b) and was conducted using a Total

Organic Carbon (TOC) analyser, Shimadzu TOC L$^{-1}$, with a TN measuring unit. Dissolved organic nitrogen (DON) was calculated by the standard approach of subtracting total inorganic nitrogen (TIN) from TN, and particular organic nitrogen (PON) as the difference between unfiltered and filtered TN. Consequently, DON and PON were negative in some years and these were subsequently set to zero. For $N_2O$-N analysis, samples were taken once or twice monthly from the drain pipe intercepting the regulation well with an air tight pumping system transferring the water to a 10 mL vial with 50 % ZnCl (0.3

25 mL per 10 mL). The $N_2O$-N samples dissolved in water were analysed using a headspace analysis technique, the head space gas in the vials being analysed on a gas chromatograph, Agilent 7890A with a CTC Combi Pal auto sampler. Total phosphorus (TP) and Orto-phosphate ($PO_4^{3-}$) were analysed according to the ISO 6878 (d.l. 0.0005 mg L$^{-1}$), and the spectrophotometric analysis was conducted on a Shimadzu UV-1700. Intensive sampling involving taking of hourly samples with an ISCO sampler was conducted for a one-week period after the regulation well opening (03/11/14) in Y1 to investigate

if the opening would lead to increased release of nutrient enriched water.

### 2.4 Isotope analysis of $\delta^{15}N$ and $\delta^{18}O$ in nitrate in drain water

Five and six grab samples of drain water from each plot were selected from Y1 and Y2, respectively, to represent scenarios with low and high water flow. Additionally, four samples from IP1 taken in Y0 were analyzed. The isotope analyses were conducted at the Stable Isotope Facility at the University of California, Davis, using the denitrifier method (d.l. of 0.4‰

for $^{15}N$ and 0.5 ‰ for $^{18}O$ (Casciotti et al., 2002;Sigman et al., 2001;Facility, 2015)). Isotope ratios of $^{15}N$ and $^{18}O$ were measured using a ThermoFinnigan Gas Bench and a PreCon trace gas concentration system interfaced to a ThermoScientific Delta V Plus isotope-ratio mass spectrometer. The international reference standards USGS 32, 34, and 35 supplied by National Institute of Standards and Technology, Gaithersburg, MD (NIST), were used to calibrate the results. Values were reported in parts per thousand (‰) relative to atmospheric $N_2$ and Vienna Standard Mean Ocean Water (VSMOW) for $\delta^{15}N$

and $\delta^{18}O$, as shown in Eq. (1):

$$\delta^{15}N \ (or \ \delta^{18}O)_{sample} \ (‰) = (R_{sample} - R_{standard}) / R_{standard} \times 1000$$




where R is the ratio of the heavy to light isotope (e.g. $^{15}N/^{14}N$ or $^{18}O/^{16}O$).

Linear regression analysis of $\delta^{18}O$ and $\delta^{15}N$ was conducted for each plot and year. The analysis of $\delta^{18}O$ vs. $\delta^{15}N$ was supplemented with the natural abundance method where $NO_3^-$-N and $\delta^{15}N$ are compared, as due to the complexity of the N cycle and the different transformation processes, the isotope compositions of $\delta^{18}O$ and $\delta^{15}N$ could be a result of the mixing of waters with different isotopic compositions rather than fractionation due to denitrification (Kendall et al., 2007).

### 2.5 Statistical and data analysis

The total loads of chemicals and water from the drainage systems were determined by multiplying the daily drain water flow by daily concentrations which were found by linear interpolation of the measured weekly concentrations. The same method of calculation was used to determine $NO_3^-$-N loss, thus modelled daily percolation and interpolation of measured concentrations in the upper groundwater was used.

A before-after control-impact design (BACI) was used to determine if controlled drainage had a significant impact on drain flow, groundwater levels and water quality (Stewart-Oaten et al., 1986). The statistical model for the BACI analysis was fitted using a repeated measures generalized linear mixed model in SAS-STAT 9.4. Data were log transformed before analysis and drain flow was transformed by log(y+1) due to the presence of null values (Box and Cox, 1964). The BACI effect (BE) was calculated as in Eq. (2):

$$BE_i = (X_{IAi} - X_{IBi}) - (X_{CAi} - X_{CBi}) \qquad \text{Paired Control and Impact Plots } (i = 1,2,..., n)$$

where $BE_i$=the BACI Effect, X=the constituent being analyzed, C=controlled, B=before, A=after and I= impacted.

To calculate percentage changes in drain flow and chemical losses data from the reference year was used to establish a ratio between plots before treatment as in Clausen and Spooner (1993). This relationship was used to predict drain flow and chemical losses at IP1 and IP2 in Y1 and Y2 and thereby the percentage changes. In Y0 N concentrations at CP2 was low due to winter wheat was grown at this plot in 2011/12, while spring barley being grown at the other plots, thus TN, $NO_3^-$-N, $NH_4^+$-N, and $N_2O$-N from CP1 was used instead in Y0.

## 3 Results

### 3.1 Hydrology

The annual precipitation in Y0, Y1 and Y2 were 783, 680 and 823 mm year$^{-1}$ and deviated by +7, -7, and +12 % from the 25-year average (1989-2014) of 733±8 mm year$^{-1}$ (Table 1). The continuously monitoring of groundwater levels showed that they fluctuated in response to interactions of precipitation, drain flow, evaporation, and percolation. Thus, the groundwater levels were higher and more fluctuating in Y2 due to a higher amount of precipitation and more evenly distributed precipitation events this year than in the preceding years. Both the spatially and continuously monitoring of groundwater levels showed that they were naturally shallow and above the drainage depth the majority of the time during the study periods (Fig. 2b).

The implementation of controlled drainage significantly elevated the groundwater table at IP1 in both Y1 and Y2 (Table 1). However, in Y1 the groundwater levels were only significantly elevated after the regulation level was increased to 70 cm on 01/28/14 (Fig. 2b). The difference between the groundwater tables at IP1 and CP1 was more profound during Y2 and, on average, 5 cm day$^{-1}$ greater at IP1 than at CP1, although varying from 0-39 cm (Table 2). The greatest differences were seen in dry periods, while during precipitation events (>10mm) the groundwater tables at CP1 reached the same level as IP1. The spatial monitoring of groundwater levels showed that the groundwater levels within 20 m from the collecting pipe (most eastern pipe) were significantly elevated during Y1 and Y2 at IP1 and IP2 (Table S3), and the extent of the area with significantly elevated groundwater levels being greater in Y2 than in Y1 now including the most southern part of IP1





and IP2. Despite elevated groundwater levels the harvest yields at the impacted plots were unaffected in both Y1 and Y2. Groundwater storage change was positive during all study periods, and no difference were seen between impacted and control plots.

The drain flow tended to follow the trends in precipitation during all years (Fig. 2a and S7). Thus, the drain flow was similar in magnitude and duration at all plots in the reference year, reflecting the high degree of similarity. The drain flow was significant reduced in both Y1 and Y2 (Table 2), and, on average, reduced by 0.18 and 0.89 mm day$^{-1}$, respectively, according to the BE. Besides drain flow rates being markedly lower at impacted plots than at controls in Y2 the drain flow ceased during periods with low precipitation (Fig. 2a). However, during precipitation events the drain flow peaks at the impacted plots were nearly at the same level as at the control plots. Additionally, drain flow occurred later at the impacted plots in Y2. The total drain flow volume from the impacted plots was reduced by 12 % and 6 % (19 and 9 mm) at IP1 and IP2 during Y1, while during Y2 reduction of 38 % and 52 % (97-127 mm) were reached at IP1 and IP2 compared to control plots, respectively (Table 1). Upon opening of the regulation well in Y1, the drain flow was 1.1 and 1.5 mm week$^{-1}$ greater from IP1 and IP2 compared with the mean drain flow from the control plots.

Vertical percolation modelled with Daisy showed that percolation accounted for 12-15 % of net precipitation in Y0 (Table 1). Since the model simulates water dynamics in a one-dimensional vertical soil column, lateral flow is not considered. The BACI analysis showed that vertical deep percolation to groundwater increased significantly at the impacted plots in both years with controlled drainage (Table 2). The daily percolation increased by, on average, 0.09 and 0.79 mm day$^{-1}$ in Y1 and Y2, respectively. This corresponded to an increase of 26-59 % and 144-309 % at the impacted plots relative to the control plots during Y1 and Y2 (Table 1).

## 3.2 Nitrogen concentrations and losses

The mean $NO_3^-$-N concentrations ranged from 5.5±0.2 to 8.3±0.3 mg L$^{-1}$ in Y0 (Fig. 2c and S9) and in general the $NO_3^-$-N concentrations varied insignificantly during the periods. The concentration at plot CP2 was considerably lower than at the other plots, probably due to the fact that winter wheat was grown at this plot in 2011/12, while spring barley being grown at the other plots (Fig. S9). The $NO_3^-$-N concentrations were significantly lower at impacted plots compared with the control plots in both Y1 and Y2; however, BE was greatest in Y1 where the $NO_3^-$-N concentrations at the impacted plots were, on average, 1.4 mg L$^{-1}$ day$^{-1}$ lower, while being, on average, 0.8 mg L$^{-1}$ day$^{-1}$ lower during Y2 (Table 2).

The loss of $NO_3^-$-N via drain flow from IP1 was reduced by 17 % (2 kg $NO_3^-$-N ha$^{-1}$) while it increased by 12 % (1 kg $NO_3^-$-N ha$^{-1}$) at IP2 during Y1, where the regulation level was kept at 50 cm above drain depth (Table 3). However, during Y2 the $NO_3^-$-N loss decreased by 36 % and 48 % (6-8 kg $NO_3^-$-N ha$^{-1}$) at IP1 and IP2. $NO_3^-$-N constituted the major part of TN loss in drainage water outflow and accounted for 95-98 % of TN in the reference year, while being lower (84-93 %) in the years with controlled drainage due to an increasing DON fraction (Table 3). Nevertheless there was no significant difference between impacted and control plots with respect to DON losses. $NH_4^+$-N loss constituted only a minimal part of the TN loss at all plots in all years (Table 3), and $NH_4^+$-N concentrations were not significantly affected by controlled drainage (Table 2, Fig. S2 and S10).The opening of the regulation well resulted in a higher loss of $NO_3^-$-N, albeit only by 26 and 93 g ha$^{-1}$ week$^{-1}$ from IP1 and IP2, respectively, compared with the mean loss from the control plots.

The $N_2O$-N concentrations were generally low and fluctuated only slightly during the study periods (Fig. S3 and S11). Mean $N_2O$-N concentrations in drain water ranged from 0.001±0.004 to 0.004±0.001 mg L$^{-1}$ during Y0. The $N_2O$-N concentrations increased significantly in Y1, while the increase at the impacted plots being insignificant during Y2 (Table 2). The indirect emission of $N_2O$-N was 5-14 g $N_2O$-N ha$^{-1}$ (51-75%) higher at the impacted plots during Y1, while it was 22-24 g $N_2O$-N ha$^{-1}$ (36-38 %) lower in Y2. The $NO_3^-$-N loss to deeper groundwater showed an increase amounting to 57 and 103 % at IP1 and IP2 during Y1 and to 171 and 393 % at IP1 and IP2 during Y2 (Table 3).





### 3.3 Total phosphorus and phosphorus concentrations and losses

The TP concentrations were generally low with means ranging from $0.008\pm0.004$ to $0.016\pm0.01$ mg $L^{-1}$ in Y0 (Fig. S4 and S12). TP increased significantly in Y1, on average, 0.003 mg $L^{-1}$ $day^{-1}$ according to the BE, but showed no significant increasing trend in Y2. In Y1, total loss of TP increased by 1.2-1.7 g (3-55 %) at the plots with controlled drainage, and in

Y2 the loss was 24.3-27.4 g (44-45 %) lower compared with the control plot losses.

The $PO_4^{3-}$ concentrations were also low with means ranging from $0.005\pm0.003$ to $0.008\pm0.004$ mg $L^{-1}$ in Y0 (Fig. S5 and S13). The BACI analysis showed that concentrations were not significantly affected in Y1 and Y2 (Table 2). The total loss of $PO_4^{3-}$ decreased by 1.3 g (21 %) at IP1 and increased by 0.8 g (12 %) at IP2 during Y1. However, during Y2 the $PO_4^{3-}$ loss was reduced by 10 and 9.6 g (54 and 44 %) at IP1 and IP2, respectively. Following the opening of the regulation

well, the loss of $PO_4^{3-}$ was 0.04 and 0.03 g $ha^{-1}$ week-1 greater at IP1 and IP2 compared with the mean $PO_4^{3-}$ loss at the control plots.

### 3.4 Isotope analysis of $\delta^{18}O$ and $\delta^{15}N$

The values of $\delta^{15}N$ and $\delta^{18}O$ indicated that $NO_3^--N$ originated from nitrification and application of organic N (Kendall et al., 2007) (Fig. S14), which is in agreement with that the potential sources of $NO_3^--N$ in this study included manure, fertilizer,

soil organic matter and wet deposition. The $\delta^{15}N$ values were within the same range (7.2-9.9 ‰) at the impacted and control plots during the two years with controlled drainage (Y1 and Y2), except for one enriched value of 13.8‰ recorded at IP1 on 02/04/14 in Y1.On this date, also the $\delta^{18}O$ value was enriched (4.3 ‰). When excluding this measurement, the $\delta^{18}O$ values were within the same range at the impacted sites and at the control plots in Y1 (0.7-2.6 ‰). In Y2, the $\delta^{18}O$ values were more or less within the same range (1.2-4.4 ‰) at impacted and control plots and thus slightly more enriched than in Y1.

The relation of $\delta^{18}O$ vs. $\delta^{15}N$ at IP1 in Y1 was within the range indicating denitrification; however, without the particularly enriched measurement from 02/04/14, there was no sign of denitrification at IP1 (slope 0.13, $R^2= 0.02$, p=0.85) (Fig. 3). For IP2, the resulting slope for $\delta^{18}O$ vs. $\delta^{15}N$ was within the excepted range for denitrification, but the relation was not significant. In Y2, the slope for $\delta^{18}O$ vs. $\delta^{15}N$ at both IP1 and IP2 suggested that denitrification occurred (Fig. 3). At CP1, denitrification seemed to be the dominant fractionation process during both years with controlled drainage. At CP2,

there was no indication of denitrification as the slope was either slightly below or slightly above the expected range.

## 4. Discussion

### 4.1 Impact of controlled drainage on hydrology

The groundwater levels were naturally shallow at all plots, thus to produce a significant groundwater level increase a regulation level of 70 cm was required, however it only resulted in an average increase of 5 cm at IP1 compared to CP1

according to the BE. Yet, it cannot be excluded that the effect of controlled drainage potentially was greater above the tile drain pipes as the monitoring of the groundwater table was conducted between pipes, the effect of controlled drainage expectedly being strongest just above the pipes (Twitty and Rice, 2001).

The spatial monitoring of the groundwater levels demonstrated that the area influenced by controlled drainage increased with the regulation level increase from 50 to 70 cm. Compared to the continuously monitored groundwater levels

the spatially monitoring of groundwater levels showed a somewhat stronger effect on groundwater levels, on average, 6 and 18 cm greater in Y1 and Y2, respectively, than at the control plots according to the BE. An explanation could be related to the timing of sampling as the difference between groundwater levels of impacted and control plots was greater in relatively dry periods.

The drain flow reduction of 6-52 % found in this study is in good agreement with findings in other field studies

conducted on silty and clayey loam; where drain flow reductions ranged between 8-64 % (Fang et al., 2012;Sunohara et al.,



2014;Tan et al., 1999;Williams et al., 2015;Drury et al., 2009;Gaynor et al., 2002). The effect of controlled drainage was clearly stronger with the highest regulation level, which cooperates with the results observed in earlier studies (Lalonde et al., 1996;Wesström and Messing, 2007;Evans et al., 1995). Besides regulation level, climate has a strong impact on the effect of controlled drainage, where the effect being smaller or even totally absent in wet years (Osmond et al., 2002). The study by Wesström and Messing (2007) showed that during wet years the effect of the highest regulation level was of the same magnitude as the effect of the lowest regulation level in dry years. During precipitation events, the drain flow peaks and groundwater table at the impacted plots were at the same level or even exceeded the peaks at the control plots. Beside reducing the drain flow, controlled drainage also resulted in an approximately one-month delay in drain flow compared with the control plots, a similar result being found by Ramoska et al. (2011). Furthermore, the drain flow ceased periodically, which might increase the soil water retention time and improve conditions for denitrification.

The water not leaving the field as drain flow at the impacted plots can either lost as evaporation, surface runoff, storage, lateral subsurface flow, vertical seepage and/or groundwater storage (Skaggs et al., 2010;Wesström et al., 2001). The pathways depend on climate, soil properties, site conditions, drainage system design and management, and crops (Skaggs et al., 2010). Only a few studies have investigated how controlled drainage alters the water pathways (Sunohara et al., 2014;Rozemeijer et al., 2015). In our study, no increase in evaporation was expected which was supported by the model simulations (Table 1). No surface runoff was observed during the study; thus, lateral subsurface flow or vertical seepage and/or groundwater storage was increased. The Daisy model simulations showed that vertical flow increased by13-237 %. However, future studies of controlled drainage should consider including a 3D hydrological model and tracer additions to cope with changes in the lateral flow component. Sunohara et al. (2014), found that lateral flow to the ditch and adjacent field increased by 61±104 % and 213±27 %, respectively, and vertical flow increased by 49±63 % on study fields compared with control fields when using the model DRAINMOD.

**4.2 Impact of controlled drainage on nitrogen fluxes**

$NO_3^-$-N concentrations were significantly reduced in both years with controlled drainage, the effect being strongest in Y1 probably driven by that $NO_3^-$-N concentrations were markedly reduced during dry periods in the beginning of February and March 2014 after the regulation level was increased with 20 cm at IP1. This was supported by that the total loading of $NO_3^-$-N declined by 22 % at IP1 while it increased by 5 % at IP2 during Y1. Nevertheless the reduction of $NO_3^-$-N loss was greatest (36-48 %) in Y2 indicating that the primary effect of controlled drainage with respect to reducing the $NO_3^-$-N loss was mediated by the reduced drain flow, which corroborate with the vast majority of studies on controlled drainage (Skaggs et al., 2012).

There was no significant difference between impacted and control plots with respect to DON losses thus no indication that controlled drainage influences the decomposition processes of organic matter releasing DON. An increase in total DON loss was seen in Y2 at all plots, but an equivalent increase in PON was not observed, which might be due to the higher groundwater levels this year, preventing macro pore flow to the drains. However, the DON and PON fractions are somewhat uncertain as they were estimated based on TN, $NO_3^-$-N or $NH_4^+$-N measurements; thus, small measurement errors may have yielded large errors in the calculated DON and PON losses (Graeber et al., 2012).

A potential disadvantage of using controlled drainage is the risk of increasing the indirect emission of $N_2O$-N via drain water, a pathway that has not been investigated in earlier studies of controlled drainage. The indirect emissions were higher in Y1 by 5-14 g $N_2O$-N ha$^{-1}$, contrary to Y2 where it was reduced by 22-24 g $N_2O$-N ha$^{-1}$ (36-38 %) due to the reduction of drain flow from the impacted plots.





### 4.3 Impact of controlled drainage on phosphorus

Even though the TP and $PO_4^{3-}$ loss was elevated in Y1 at IP2, it was only by 0.8 and 1.7 g bearing in mind that the impact of controlled drainage at this site was minimal in Y1. In Y2 when controlled drainage was effective, both $PO_4^{3-}$ and TP loss declined, since the concentrations were not significantly affected, indicating that the reduction was a consequence of the reduced drain flow. The findings that $PO_4^{3-}$ concentrations was unaffected by the implementation of controlled drainage could be explained by that groundwater level was not elevated long enough to create reduced conditions to the levels where $PO_4^{3-}$ is released as suggested in Williams et al. (2015). Furthermore, the $PO_4^{3-}$ loss did not pose a risk when opening the regulating well in this setup, as the loss was imperceptible small.

### 4.4 Impact of controlled drainage on denitrification

The results of Y1 showed that on 02/04/14, enhanced denitrification occurred at IP1, $\delta^{15}N$ and $\delta^{18}O$ being enriched by 2:1 manner with a pronounced reduction of $NO_3^-$-N, while $NH_4^+$-N and $SO_4^{2-}$ increased (Fig. S2 and S6). The increase of $NH_4^+$-N and $SO_4^{2-}$ concentrations indicated that both heterotrophic and autotrophic denitrification occurred. A natural source of $SO_4^{2-}$ can be omitted since $SO_4^{2-}$ levels did not increase at the other plots, and no fertilizer was applied during the winter monitoring periods. Furthermore, the significantly higher $N_2O$-N concentrations in drainage water at the impacted plots than at control plots suggest denitrification. To investigate if IP1 was particularly prone to denitrification due to the high C content in the deeper soil horizons, stable isotope analyses of $NO_3^-$-N in drain water from Y0 were conducted. The slope of $\delta^{18}O$ vs. $\delta^{15}N$ at 2.01 ($R^2= 0.94$, $p<0.005$) suggested that denitrification did not take place in Y0, and no reduction in $NO_3^-$-N concentrations were seen. Thus, the enhanced denitrification found during Y1 could be related to the changes in hydrology, including cease of drain flow and stable groundwater levels above drain depth, leading to an increase of the soil water retention time. The results from IP2 in Y1 were more ambiguous as the relationship between $\delta^{18}O$ and $\delta^{15}N$ was within the range of denitrification but insignificant, potentially due to the low sampling frequency. Nevertheless, the negative correlation between $\delta^{15}N$ and $NO_3^-$-N indicated that denitrification took place.

The regression analysis of $\delta^{15}N$ and $\delta^{18}O$ indicated that denitrification occurred at the impacted plots when the regulation level was 70 cm above drain depth in Y2. This was supported by a strong negative correlation between $\delta^{15}N$ and $NO_3^-$-N ($R^2= 0.65-0.74$) in at both impacted plots. The $N_2O$-N concentrations were also found to increase, albeit not significantly so, at the impacted plots. According to the stable isotope analysis, denitrification also took place at the control plots in both Y1 and Y2, especially at CP1. At CP2, the result was slightly out of range, but the relationship of $NO_3^-$-N and $\delta^{15}N$ indicated that denitrification did occur. This is in agreement with the expectation that denitrification would occur at all sites, as predicted by the simple model SimDen (Vinther and Hansen, 2004), which estimated a yearly denitrification potential of 18-23 kg N $ha^{-1}$ $yr^{-1}$ in this soil type with the given amount of applied manure and fertilizer.

The application of stable isotope analysis of $NO_3^-$-N to trace denitrification is far from trivial due to the complexity of the N cycle (Kendall et al., 2007). The results demonstrated markedly enhanced denitrification only on 02/04/14 at IP1. The reason for the absence of a distinct increase in denitrification may owe to several factors. Firstly, the impact of controlled drainage on groundwater levels appeared to be lower than expected. Secondly, the soil water might not be adequate anoxic for denitrification to occur as percolating water from the soil surface will supply oxygen ($O_2$) to the soil water.

### 5 Conclusion

The introduction of controlled drainage on loamy soils with winter crops did significantly change tile drain flow and nitrogen loss, however, only when a regulation level of 70 cm above drain depth was applied. The drain water flow was reduced by 38-52 % (97-127 mm) and a nearly similar reduction was found for the $NO_3^-$-N loss via drain pipes, which was reduced by





36-48 %, corresponding to 6-8 kg $NO_3^-$-N per ha. In this study controlled drainage did not lead to 'pollution swapping' where reduction in N loss is substituted by increased emission of nitrous oxide or dissolved reactive phosphorus, which has been shown to be a potential risk when applying controlled drainage. Markedly enhanced denitrification was only documented for a shorter, dry period in one of the controlled drainage systems, as shown by the results of a stable isotopes analysis supported by strongly declining $NO_3^-$-N concentrations and an increase in concentrations of $NH_4^+$-N and $SO_4^{2-}$.

Controlled drainage can only be considered an effective mitigation measure if the overall loss of $NO_3^-$-N to groundwater and surface water is reduced. However, the results from this study do not allow us to determine the pathways for the groundwater and $NO_3^-$-N that under controlled drainage did not enter. Therefore, it is an open question if the missing $NO_3^-$-N in tile drains on fields with controlled drainage will be reduced on its pathway to surface waters. Therefore, knowledge about how controlled drainage changes water flow pathways and the likelihood of $NO_3^-$-N being reduced is necessary to elucidate the net effect of controlled drainage on the $NO_3^-$-N loss.

*Acknowledgement.* The authors are grateful for assistance from the technicians at Aarhus University. The research project was financed by 'Green Development and Demonstration Programme (GUDP)' of The Danish Ministry of Environment and Food.

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

20

25





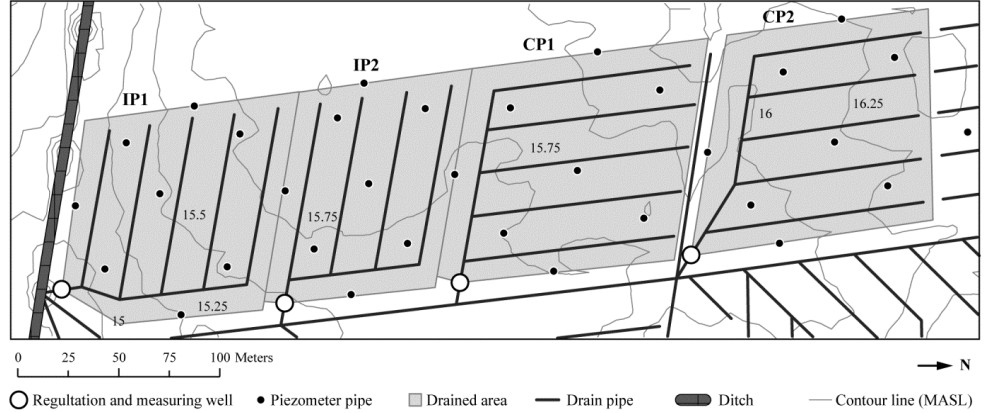

**Figure 1.** Map of the experimental site.



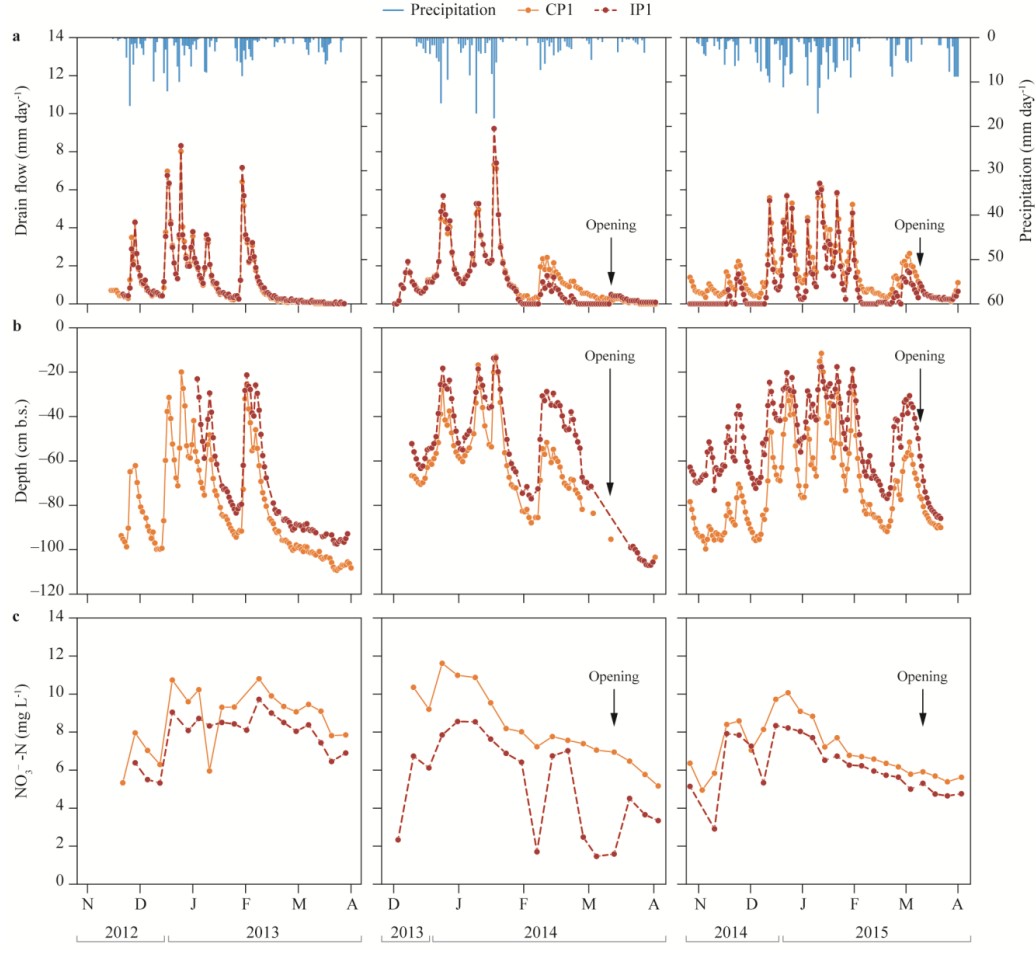

**Figure 2.** Drain flow (a), groundwater levels (b), and NO$_3^-$-N concentrations (c) at impacted (IP1) and control plot (CP1) during Y0 (2012/13), Y1 (2013/14), and Y2 (2014/15).





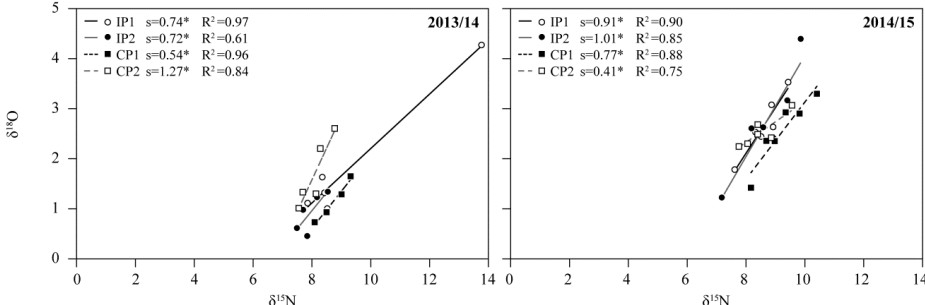

**Figure 3.** $\delta^{18}O$ vs. $\delta^{15}N$ at impacted (IP1-2) and control plots (CP1-2) during Y1 (2013/14) and Y2 (2014/15).

20




**Table 1**. Water balances for the period Y0-2 at impacted (IP1-2) and control plots (CP1-2).

| | Y0 (2012/13) | | | | Y1 (2013/14) | | | | Y2 (2014/15) | | | |
|---|---|---|---|---|---|---|---|---|---|---|---|---|
| | IP1 | IP2 | CP1 | CP2 | IP1 | IP2 | CP1 | CP2 | IP1 | IP2 | CP1 | CP2 |
| Precipitation | 262 | 262 | 262 | 262 | 284 | 284 | 284 | 284 | 350 | 350 | 350 | 350 |
| AE | -44 | -44 | -44 | -53 | -69 | -69 | -69 | -69 | -57 | -57 | -55 | -55 |
| Drain flow | -166 | -157 | -159 | -169 | -139 | -140 | -152 | -160 | -160 | -116 | -255 | -253 |
| Matrix percolation | -46 | -40 | -47 | -40 | -49 | -54 | -43 | -32 | -119 | -173 | -49 | -43 |
| ΔGW | 34 | 36 | 30 | 20 | 5 | 12 | 26 | 4 | 27 | 13 | 18 | 7 |
| Net inflow | -40 | -57 | -42 | -20 | -32 | -33 | -46 | -27 | -41 | -17 | -9 | -6 |





**Table 2**. BACI test for the periods Y0 (2012/13), Y1 (2013/14) and Y2 (2014/15) for the impacted (IP1-2) and control plots (CP1-2).

|  | Unit of BE | Compared periods | n* | BE | t value | p |
|---|---|---|---|---|---|---|
| Drain flow | mm day$^{-1}$ | Y0/Y1 | 976 | 0.2 | 13.6 | <0.001 |
|  | - | Y0/Y2 | 1024 | 0.9 | 46.2 | <0.001 |
| Groundwater level | cm day$^{-1}$ | Y0/Y1 | 353 | -0.1 | 2.9 | <0.001 |
|  | - | Y0/Y2 | 426 | -4.9 | 5.1 | <0.001 |
| Percolation | mm day$^{-1}$ | Y0/Y1 | 976 | -0.1 | -16.1 | <0.001 |
|  | - | Y0/Y2 | 1024 | -0.8 | -47.0 | <0.001 |
| TN | mg L$^{-1}$ day$^{-1}$ | Y0/Y1 | 148 | 1.5 | 4.5 | <0.001 |
|  | - | Y0/Y2 | 156 | 0.7 | 3.3 | <0.001 |
| NO$_3^-$-N | mg L$^{-1}$ day$^{-1}$ | Y0/Y1 | 148 | 1.4 | 4.3 | <0.001 |
|  | - | Y0/Y2 | 156 | 0.8 | 3.9 | <0.001 |
| NH$_4^+$-N | mg L$^{-1}$ day$^{-1}$ | Y0/Y1 | 148 | -0.010 | -0.1 | 0.91 |
|  | - | Y0/Y2 | 156 | -0.001 | 1.6 | 0.12 |
| N$_2$O-N | µg L$^{-1}$ day$^{-1}$ | Y0/Y1 | 36 | -7.0 | -3.06 | <0.001 |
|  | - | Y0/Y2 | 28 | -5.3 | -0.01 | 0.99 |
| TP | mg L$^{-1}$ day$^{-1}$ | Y0/Y1 | 148 | -0.003 | -3.1 | <0.05 |
|  | - | Y0/Y2 | 156 | b.d.l. | -1.3 | 0.19 |
| PO$_4^{3-}$ | mg L$^{-1}$ day$^{-1}$ | Y0/Y1 | 148 | b.d.l. | -1.0 | 0.31 |
|  | - | Y0/Y2 | 156 | b.d.l. | 0.1 | 0.88 |

*sample size





**Table 3.** Total losses of chemicals via drainage water and groundwater during Y0-2 at impacted (IP1-2) and control plots (CP1-2).

| | Y0 (2012/13) | | | | Y1 (2013/14) | | | | Y2 (2014/15) | | | |
| --- | --- | --- | --- | --- | --- | --- | --- | --- | --- | --- | --- | --- |
| | IP1 | IP2 | CP1 | CP2 | IP1 | IP2 | CP1 | CP2 | IP1 | IP2 | CP1 | CP2 |
| TN | 14.0 | 12.8 | 15.1 | 11.1* | 10.9 | 14.3 | 16.5 | 13.4 | 13.0 | 9.9 | 22.6 | 20.3 |
| $NO_3^-$-N | 13.5 | 12.6 | 14.8 | 10.5* | 10.1 | 12.7 | 14.3 | 12.4 | 11.0 | 8.4 | 19.7 | 20.3 |
| $NH_4^+$-N | 0.013 | 0.026 | 0.026 | 0.028* | 0.010 | 0.005 | 0.006 | 0.006 | 0.009 | 0.006 | 0.017 | 0.013 |
| PON | 0.25 | 0.19 | 0.78 | 0.27* | 0.06 | 0** | 0.14 | 0.11 | 0.21 | 0** | 0.18 | 0.18 |
| DON | 0.16 | 0.01 | 0** | 0.25* | 0.67 | 1.60 | 2.04 | 0.89 | 1.74 | 1.54 | 2.68 | 2.13 |
| $N_2O$-N | 0.026 | 0.014 | 0.013 | 0.007* | 0.034 | 0.015 | 0.012 | 0.008 | 0.035 | 0.019 | 0.030 | 0.025 |
| TP | 0.044 | 0.013 | 0.023 | 0.046 | 0.009 | 0.004 | 0.006 | 0.006 | 0.046 | 0.013 | 0.053 | 0.061 |
| $PO_4^{3-}$ | 0.016 | 0.009 | 0.011 | 0.019* | 0.006 | 0.005 | 0.007 | 0.008 | 0.013 | 0.006 | 0.017 | 0.027 |
| $NO_3^-$-N via groundwater | 3.4 | 2.8 | 3.4 | 2.6 | 8.1 | 8.7 | 6.0 | 4.6 | 12.5 | 18.9 | 5.3 | 4.1 |

* values from CP1 was used in the calculations,**corrected to zero.

