# Peer review of "Can controlled drainage control agricultural nutrient emissions? Evidence from a BACI experiment combined with a dual isotope"

_Hydrology and Earth System Sciences, 2016_

## Referee Comment (RC1) · Anonymous Referee #1 · 19 Aug 2016

General comments

The authors tested the impact of controlled drainage (maintaining a higher water table) on subsurface N losses from soils. The work was carried out in a multi-year replicated field, where the water table was kept artificially high in two of the four plots for the final two years. Once groundwater tables increased, both NO3- concentrations in the drainage water and the drainage discharge flow decreased. In order to test whether changes were caused by enhanced denitrification, the authors measured dissolved N2O and NO3- isotopic composition in the drainage water. The overall scope of the study is robust, and definitely of interest. However, lack of information in the methods section and inconsistencies in the presentation of the data call into question the validity

of this work. I would also recommend careful proof-reading and reorganisation of the manuscript, as the meaning was often difficult to follow.

Specific comments

Material & Methods

• More information is needed on the management of the experimental plots particulary on the activities known to alter N leaching. This includes quantity, timing, and quality of fertilisers, as well as ploughing, fallowing, crop choice, and crop yields. In the first paragraph of the M&M it says that, "Field management practices were similar during the three-year study period involving growth of winter wheat and application of identical amounts of manure and fertilizer in the spring." Yet the last line of section 2.5 then says that differences between the plots were due to some being planted with winter wheat and some with barley. In section 2.2 it says that field management data was obtained, and in section 2.1 it says that harvest data was also obtained, yet none of this information is presented in the manuscript. This information needs to be established in order to interpret temporal trends in N losses, as well as differences between the control and treatment plots. I suggest adding a schematic timeline of the management scheme to Fig. 1, as well as including indicators of key events such as fertiliser application and implementation of controlled drainage to Fig. 2.

• There are a couple of caveats to the overarching experimental design that are not explained clearly: 1) tweaking of the water table level in the treatments plots (was the procedure identical in both treatment plots? The date also isn't clear. This should be included both in a field management timeline and indicated on the figures showing changes in the water table over time), and, 2) opening the outflow gate (did this only happen in one of the treatment plots? In the methods section it says that intensive samples were collected during this period, but the data is not shown or discussed). As these events aren't well explained, they do call into question how representative the overall findings are.
Data quality

• How were the water samples preserved prior to analysis? The lack of mention of any filtration, freezing, etc. make it seem likely that the reported N concentrations and isotopic compositions do not represent the field conditions.

• The first sentence of section 2.5 states that yearly loads were calculated by first dividing the weekly measured nutrient concentrations into daily fluxes via linear interpolation over time. This approach assumes a constant relationship between nutrient export and time. However, this assumption is not consistent with previous findings that, e.g., nitrate concentrations tend to decrease with increased flow. It would therefore be most accurate to calculate total loads based only on the days when stream chemistry data was collected.

• In Table 3 it says that the Y0 column for CP2 is actually filled with values for CP1. What happened to the CP2 data? Why was it excluded? If the data from CP2 was unusable, then this should simply not be included in the table, and a statement about why the data was excluded added to either the results or the M&M. Filling this column with data from the other control plots is misleading, at best.

Data presentation

The data presentation seems overly selective, making it difficult to follow the results or ascertain the accuracy of the conclusions. Most critically:

• Figure 2 only shows data over time for two of the four plots. The other two need to be included if data from them is going to be discussed. The decision to separate each year into a unique (yet unlabelled) sub-plot also makes this figure hard to follow. I'd recommend plotting data from all four plots over a continuous x-axes, using arrows, lines, or shading to indicate the periods that correspond with the 'y0', 'y1', and 'y2' referred to in the text.

• Figure 3: This figure only shows data from Y1 and Y2. Where is the Y0 data?

Additionally, the meaning of the astricts adjacent to the r2 values listed within the plate are not explained in the figure caption, and the slopes reported here do not seem to correspond with those mentioned in the discussion.

• Units are needed for all parameters in Table 1 and Table 3, as well as quantitative information on uncertainty for each number shown

• In the final sentence of paragraph three in section 4.1 it says that, ". . .controlled drainage also resulted in an approximately one-month delay in drain flow compared with control plots.". As drain flow shown in Fig. 1 does not seem to support this, more evidence on where this statement comes from is needed.

• Nutrient data is presented as concentrations (when units are shown), but the focus of the paper is 'loss' (i.e., concentration x discharge x time), it would therefore be useful to see the data in flux units (g s-1).

• N2O data is only shown in terms of dissolved concentrations. As water in the drainage system will be influenced by both atmospheric N2O and biogenic N2O, it would be more useful to discuss these findings in terms of % saturation. Emissions of N2O from the system also depend on saturation dynamics (see classic description of N2O solubility in Weiss & Price (1980) Marine Chemistry).

• Section 2.2 says that groundwater (∼7 piezometers per plot shown in Fig. 1) was sampled monthly for nutrient concentrations. However, the only groundwater data shown is the (unitless) annual nitrate value in Table 3. How variable were the concentrations over time? Did they differ between the control and treatment plots? How was groundwater data used to calculate N and P losses? What was the P concentration in groundwater?

• More information is need on the spatial and temporal variability in other nutrient parameters discussed (N2O, P, NH4+, SO4, DON, and PON). While some of this data is included in supplemental figures, the critical parameters should be included in the

main manuscript in order to create a coherent and convincing story. This could be as simple as adding information on variability and sample numbers to Table 3.

Data interpretation

• Given the experimental design, this paper needs to be organised to more logically explain how variables are, 1) different in treatment plots before and after induced conditions, and, 2) how treatment plots differed from control plots (i.e., where they the same prior to changed drainage conditions, as in, were the controls actually good controls?). The results and discussion are very disorganised, and the selective data displayed, make it hard to tease out the answer to either of these questions

• The discussion around the NO3- isotope data is a bit hard to follow. First, it would be useful to include a 95% CI for each slope described in Fig. 3 in order to more accurately judge if they overlap with the range expected for denitrification (1:1 – 2:1). As it seems that all of the data does plot roughly along a denitrification line, section 4.4 needs to be revised to discuss the data in terms of NO3- 'more impacted' v 'less impacted' by denitrification as values move up and down the denitrification line. It would then be useful to discuss what factors influenced these moves. As the authors note in the second paragraph of 4.4, denitrification is probably always occurring somewhere in an arable soil. It's therefore useful to keep in mind that the leached NO3- isotopes are a reflection of the degree to which denitrification is controlling the NO3- flux, and not direct measures of denitrification activity. This also means that it's a bit of an overstatement to say that higher NO3- isotopes show enhanced denitrification on a specific day. Instead, this higher value may indicate that reducing conditions dominated in the period prior to sampling (though, as this was only observed in one of the three plots, it also seems possible that this sample wasn't particularly representative of reality?). Overall the ∼1:1 ratio of d18Ovd15N suggests that NO3- leached from the plots has undergone variable degrees of denitrification. So what controls these variations? Did isotope values increase in response to rainfall, season, temperature? And are these variations different between the control and treatment plots? I suggest checking out the

recent advances in the interpretation of NO3- isotope data from, e.g., Hall et al. (2016) Oecologia and Wells et al. (2016) Water Resources Research when re-evaluating this data.

• The abstract and conclusion both mention 'pollution swapping', whereby decreases in NO3- leaching are countered by increases in N2O emissions. Here the drain N2O data is interested from the point of view of obtaining a more complete picture of N leaching losses, but not conclusive evidence for/against pollution swapping. This is because soil surfaces are the primary source of N2O emissions (and thus the focus of concern in 'pollution swapping' follow drainage manipulation). Additionally, it is unclear if / how dissolved N2O was affected by controlled drainage, as in the first paragraph of section 4.4 it says that N2O-N was higher in the impacted plots, but then in the next paragraph it says that differences in N2O-N concentrations were not significant.

• The conclusions seem to say that the manuscript makes no contribution towards understanding controlled drainage systems. A clearer case for why this manuscript should be published / read is needed.

---

## Referee Comment (RC2) · Anonymous Referee #2 · 4 Sep 2016

General comment

This paper focuses on the assessment of the impacts of controlled drainage on drain flow, groundwater levels and nutrient emission. The authors used a before-after control-impact (BACI) sampling design in four adjacent drainage systems to test whether the controlled drainage had a significant impact on nutrient losses. They found that controlled drainage significantly affects the decrease in drain water flow and nitrate loss. The authors also combined the BACI experiment with a dual isotope approach (relation between $\delta 18O$ and $\delta 15N$) to determine whether denitrification occurred in the impacted plots. The aim of the study is of interest for the readers of the journal and overall the paper is well written. Nonetheless, I suggest changes to the materials and

methods section to add some important details, which are missing, and rephrase the conclusions to better highlight the novelty of the study.

Specific comments

- The authors state that field management practices were similar during the three-year monitored period (lines 34-35, page 2), but at lines 22-24, page 5, they justify the lower nitrate concentration with the different agricultural management in the plots in 2011/2012. To avoid inconsistencies throughout the paper, I suggest to describe the field management practices carried out in the plots during the experiment (including the quantities of fertilizers) and clarify the possible effects of previous managements on the results obtained with the BACI experiment.

- In lines 28-30, page 3, the authors refer to an intensive sampling campaign carried out in Y1 to assess whether the opening would lead to an increase in the release of nutrient enriched water. The results of this intensive campaign are not reported in the paper. Do these results support the findings and are they relevant for the paper? If they are not relevant for the paper, it is better to remove the sentence in the Materials and Methods to improve the clarity of the section.

- Sections 2.3 and 2.4 omit how the water samples were stored before the analyses, if they were filtered and analysed immediately after the collection. These details should be included in the two sections.

- Figure 1 reports that there are eight piezometers installed in each plot for groundwater level measurements (and water sampling), but in Fig. 2b there are only two series of dots. Do the dots represent an average groundwater level? If so, this information should be included in the caption and the authors should discuss the spatial and temporal variability of groundwater levels and nutrient concentrations and report which values (all the data collected?) they used for the BACI test (Table 2) and for the calculations of total losses of chemicals (Table 3). Furthermore, the description of the locations of piezometers in the plot, as reported in Table S3, is quite confusing. Is it

[Figure]

possible to add letters/numbers in Fig. 1 or have another map in the supplementary material?

- The authors should explain why they replaced CP2 values with CP1 in the calculations for Table 3. Did the authors assume that the difference between the samples collected at the two control plots is not significant?

- In Section 4.4 (lines 16-18, page 8) the authors report the slope for the relation between $\delta$18O and $\delta$15N in Y0 and comment it. In order to improve the consistency and compare Y0 with Y1 and Y2, is it possible to add the data in Fig. 3?

- The Conclusions section reports briefly the main findings of the study, but the novelty is not very clear or is not highlighted as it should be. Therefore, I would recommend to rephrase the Conclusions.

Technical corrections

- Figure 2: Please report the origin of nitrate concentrations (drain water?).

- Figure 3: Measurement units are missing in the x and y axis. Please add them and zoom in to improve the readability of the figure.

- Table 1: Please add the measurement units and standard deviations whether average values are reported in the table.

- Table 2: Please report in the caption what 'b.d.l.' means.

- Table 3: Please add the measurement units and standard deviations whether average values are reported in the table.

[Figure]

---

## Referee Comment (RC3) · J. Rozemeijer (Referee) · 9 Sep 2016

This paper presents a solid study on the effects of controlled drainage on nutrient losses from an agricultural field. The paper is well written and well structured.

I have one major comment. The authors conclude that the water discharge and nitrate losses to surface water via the subsurface drainage system has considerably reduced after implementing controlled drainage. This raises the question where this water and nitrate goes instead. There was no influence on the harvest yield, so probably no extra evapotranspiration and crop uptake. Denitrification was also not markedly enhanced. The authors report that no overland flow was observed. The water and nitrate must have infiltrated to the upper groundwater. From there, the fate remains uncertain. The

extra nitrate load may have polluted the deeper groundwater resources. In this case, pollution swapping did occur; less nitrate loss to surface water, more nitrate loss to groundwater. The other option is an enhanced shallow groundwater flow towards the surface water. In this case, there is no reduction of the nitrate load to surface water.

The uncertainty about the fate of nitrate is described in the discussion (p7 L11-21) and in the conclusion (p9 L6-11). However, this crucial aspect is missing in the abstract, which only presents the positive effects of controlled drainage. In the discussion, I would expect a more thorough discussion about the potential negative effects. Furthermore, an evaluation of the research methodology could be added to the discussion. How can the total effects of controlled drainage be quantified in future studies? The authors only suggest tracer additions and 3D modelling (p7L18-19). Could more intensive hydrological and chemical monitoring of different flow routes also add to this? A sentence evaluating the monitoring setup could also be added to the abstract and the conclusions.

Minor comments: P1L12: 'For the first time': it's unclear what exactly was for the first time. A controlled drainage pilot in Denmark? A controlled drainage pilot on a field with winter crops? Controlled drainage as mitigation for nitrate losses? Etc.

P2L45: In addition to anoxic conditions, you also need organic matter or pyrite for denitrification.

P2L13-17: These 'hypotheses' are formulated here as questions. Replace hypotheses with research questions?

P4L27-30: Higher and more fluctuation groundwater levels due to more evenly distributed precipitation events?

P4L32:"The implementation. . .(Table 1 )" I don't understand how this follows from table 1.

P4L33-35: Could you add more information about the regulation level management of

the controlled drainage system? This should be part of section 2.

P4L35: 5 cm per day? Why per day?

---

## Author Comment (AC1) · 7 Oct 2016

Dear referee

We highly appreciate your constructive comments and suggestions that will surely improve our manuscript.

To answer your comments best possible we have divided them into small sections which we will respond to individually.

**Material & Methods**

A. More information is needed on the management of the experimental plots particulary on the activities known to alter N leaching. This includes quantity, timing, and quality of fertilisers, as well as ploughing, fallowing, crop choice, and crop yields. In the first paragraph of the M&M it says that, "Field management practices were similar during the three-year study period involving growth of winter wheat and application of identical amounts of manure and fertilizer in the spring." Yet the last line of section 2.5 then says that differences between the plots were due to some being planted with winter wheat and some with barley. In section 2.2 it says that field management data was obtained, and in section 2.1 it says that harvest data was also obtained, yet none of this information is presented in the manuscript. This information needs to be established in order to interpret temporal trends in N losses, as well as differences between the control and treatment plots. I suggest adding a schematic timeline of the management scheme to Fig. 1, as well as including indicators of key events such as fertilizer application and implementation of controlled drainage to Fig. 2.

We appreciate your suggestion and agree with that management data have to be more detailed. To fulfill this we will add a table (shown below) containing management practice to either the main article or to the supplementary. With respect to line 2.2 and 2.5 the management during the experiment (Y0-Y2) was similar (with minor difference in fertilizer amount), but during the year before (2011/12) the experiment spring barley was grown at IP1-1 and CP1, while at CP2 winter wheat was grown, which resulted in lower N concentrations at CP2 in the following autumn, which was our reference year.

| Plots | 2011/12 IP1, IP2, CP1 | CP2 | 2012/2013 (Y0) IP1, IP2, CP1 | CP2 | 2013/14 (Y1) IP1, IP2, CP1 | CP2 | 2014/15 (Y2) IP1, IP2, CP1 | CP2 |
|---|---|---|---|---|---|---|---|---|
| **Crop** | SB | WW | WW | WW | WW | WW | WW | WW |
| **Plowing** | 26 mar | 26 sep | 8 oct | 17 sep | 16 sep | 16 sep | | |
| **Sowing** | 27 mar | 27 sep | 9 oct | 18 sep | 17 sep | 17 sep | 18 sep | 18 sep |
| **Fertilizer application:** | | | | | | | | |
| Pig slurry | 10 may | 19 apr | 1 may | 1 may | 5 may | 5 may | 1 may | 1 may |
| -amount (ton) | 20 | 30 | 30 | 25 | 18 | 18 | 36 | 36 |
| Mineral, 1st | 27 mar | 16 mar | 15 mar | 8 apr | 26 mar | 26 mar | 20 mar | 20 mar |
| -amount (kg) | 103[a] | 156[b] | 125[a] | 125[a] | 200[c] | 200[c] | 150[d] | 125[d] |
| Mineral, 2st | | 20 apr | 20 apr | 9 may | 15 apr | 15 apr | 20 apr | 20 apr |
| -amount (kg) | | 172[b] | 194[b] | 194[b] | 215[c] | 165[c] | 100[b] | 100[b] |
| **Harvest** | 21 aug | 21 aug | 21 aug | 21 aug | 21 aug | 21 aug | 21 aug | 21 aug |
| -yield (hkg ha$^{-1}$) | 81, 81, 82 | 105 | 98, 100, 97 | 86 | 98, 99, 99 | 99 | 80, 75, 81 | 75 |

SB=spring barley, WW=winter wheat, Fertilizer type: [a]NS 21-24, [b]NS 27-4, [c]NS 28-5, [d]NS 26-13.

B. There are a couple of caveats to the overarching experimental design that are not explained clearly: 1) tweaking of the water table level in the treatments plots (was the procedure identical in both treatment plots? The date also isn't clear. This should be included both in a field management timeline and indicated on the figures showing changes in the water table over time), and, 2) opening the outflow gate (did this only happen in one of the treatment plots?

Thanks for pointing this out. The specific dates are not mentioned, but the overall periods are in section 2.1 (line 23-25), however we agree that it is not sufficient and it can be distracting not knowing the dates. In the main article we will notify that the regulation wells at IP1 and IP2 were closed in the end of October (three-five weeks before drain flow began) and the dates where the regulation wells were opened (30/11/2014 and 30/09/2015) (also shown in figure 2). Furthermore, we will add the table shown below to the supplementary information.

However, in section 2.1(line 24-28) the treatments are described. The regulation levels were similar (50 cm) in Y1 at IP1 and IP2 until the 01/28/2014, where the level was increased to 70 cm at IP1, and in Y2 both plots had a level of 70 cm.

| Plots with CD | IP1 and IP2 | |
|---|---|---|
| Plots without CD | CP1 and CP2 | |
| Management of regulation well at IP 1-2 | closed | opened |
| Y1
Y2 | 10-dec-13
17-nov-14 | 11-mar-14
09-mar-15 |
| Reference period | Y0 (21-nov-2012 to 21-apr-2013) | |
| Regulation level in
Period 2
Period 3 | 50 cm *
70 cm | |
| Number of piezometers pr. plot with pressure transducer | 1 | |
| Number of piezometers pr. plot without pressure transducer | 8 | |
| Frequency of water sampling in piezometers | 2-3 times a month | |
| Frequency of water sampling in the measuring well | Weekly | |
| Frequency of drain water flow measurement | Every 10th minute | |
| Frequency of ground water level measurements in piezometer with pressure transducer | Daily** | |
| Frequency of ground water level measurements in piezometer with continuous pressure transducer | 2-3 times a month | |

* until 28 January 2013 for CP1 hereafter 70 cm.
** Often lower frequency due to low inflow time of soil water, thus data from IP2 from all periods was unusable. Dysfunctional pressure transducer at CP1 in beginning of Y0 and at CP2 in Y3.

C. In the methods section it says that intensive samples were collected during this period, but the data is not shown or discussed. As these events aren't well explained, they do call into question how representative the overall findings are.
We are grateful that you emphasis this and the data will be either added to Fig 2 or as supplementary information. In section 4.3 (line 7-8) of the discussion the results of the intensive sampling is discussed with respect to phosphorus loss, but the information is not discussed with respect to water and nitrate loss, which will be added.

D. How were the water samples preserved prior to analysis? The lack of mention of any filtration, freezing, etc. make it seem likely that the reported N concentrations and isotopic compositions do not represent the field conditions.
Thanks for pointing this out. The information will be added to the methods.

E. The first sentence of section 2.5 states that yearly loads were calculated by first dividing the weekly measured nutrient concentrations into daily fluxes via linear interpolation over time. This approach assumes a constant relationship between nutrient export and time. However, this assumption is not consistent with previous findings that, e.g., nitrate concentrations tend to decrease with increased flow. It would therefore be most accurate to calculate total loads based only on the days when stream chemistry data was collected.
Several authors have shown that it is of greater importance to include the daily variation in drain flow which in our case was measured with flow meters, than variation in daily concentration of nutrients. We have in line with most suggestions in the international literature (cf. Kronvang and Bruhn, 1996; Grant et al., 1996) decided also to estimate a daily nutrient concentration and calculate the transport of nutrients by multiplying the measured average daily drain flow and estimated (with linear interpolation from weekly samples) daily concentrations.

Kronvang, B. and Bruhn, A.J., 1996: Choice of sampling strategy and estimation method when calculating nitrogen and phosphorus transport in small lowland streams. Hydrological Processes, Vol. 10, 1483-1501.

Grant, R., Laubel, A., Kronvang, B., Andersen, H.E., Svendsen, L.M. and Fuglsang, A., 1996: Loss of dissolved and particulate phosphorus forms in drainage water from four arable catchments on structured soils in Denmark. Water Research, 30(11), 2633-2642.

F. In Table 3 it says that the Y0 column for CP2 is actually filled with values for CP1. What happened to the CP2 data? Why was it excluded? If the data from CP2 was unusable, then this should simply not be included in the table, and a statement about why the data was excluded added to either the results or the M&M. Filling this column with data from the other control plots is misleading, at best.
Thanks to your comments we realize that the footnote of Table 3 is poorly phrased. The data represented in the table is the results from CP2, however these results were omitted from the analysis (BACI and calculation of percentage loss) as winter wheat was grown at this field prior to the experiment (2010/11), while spring barley was grown at the

other plots. The consequence was that N concentrations were much lower at CP2 compared to the other plots in 2012. Thus it was decided to omit the data from CP2 in Y0. This will be explained more clearly in the footnote.

**Data presentation**

G. The data presentation seems overly selective, making it difficult to follow the results or ascertain the accuracy of the conclusions. Most critically: Figure 2 only shows data over time for two of the four plots. The other two need ´ to be included if data from them is going to be discussed. The decision to separate each year into a unique (yet unlabelled) sub-plot also makes this figure hard to follow. I'd recommend plotting data from all four plots over a continuous x-axes, using arrows, lines, or shading to indicate the periods that correspond with the 'y0', 'y1', and 'y2' referred to in the text.
Thanks for stating your interpretation of Fig.2. We choose to show results only from two of the plots (one treated and one control) to make the interpretation of the data easier as the results were much alike and the plots got "crowded" when all four plots were shown in the figure. The results of the two other plots are presented in the supplementary information.
The subplots are labelled (just below the lowest x-axis), but it have to be made more visible if it can be overseen. We have only shown data from the study periods to emphasis the differences in these periods.

H. Figure 3: This figure only shows data from Y1 and Y2. Where is the Y0 data? ´Additionally, the meaning of the astricts adjacent to the r2 values listed within the plate are not explained in the figure caption, and the slopes reported here do not seem to correspond with those mentioned in the discussion.
We appreciate that you have noticed that the explanation of the astricts is lacking, which will be added (the astricts indicate that the relation is significant). Data from Y0 were not shown on purpose in Fig. 3, as we only have data from IP1 (it was a more targeted investigation where we wanted to find out if denitrification was more likely to occur at this plot, due to the high carbon content in the lower root zone), so instead the result was addressed in the discussion. However we will consider adding the data from Y0 and confidence intervals to the Fig.3 to make the interpretation easier, also the results from IP1 in Y0 should be addressed in the results.
None of the slopes shown in the Fig.3 are mentioned in the discussion, as they are shown in the figure. The only slope mentioned in the discussion is the slope of the data from Y0 (which is the data not shown in Fig. 3).

I. Units are needed for all parameters in Table 1 and Table 3, as well as quantitative ´ information on uncertainty for each number shown
We appreciate your observation and apologies the inconvenience. Units will be added (the unit of table 1 is mm and the unit of table 3 is kg ha$^{-1}$).

J. In the final sentence of paragraph three in section 4.1 it says that, " ´ . . .controlled drainage also resulted in an approximately one-month delay in drain flow compared with control plots.". As drain flow shown in Fig. 1 does not seem to support this, more evidence on where this statement comes from is needed.

The delay can be seen in Fig. 2a (upper-right), however this might be difficult to interpret due to the easily overlooked labels of the figure, which will be changes (see answer G). It is easier to see if the figure is viewed in colour.

K.  Nutrient data is presented as concentrations (when units are shown), but the focus ´ of the paper is 'loss' (i.e., concentration x discharge x time), it would therefore be useful to see the data in flux units (g s-1).
The unit of nutrient data shown in Table 3 is kg ha$^{-1}$.

L.  N2O data is only shown in terms of dissolved concentrations. As water in the drainage system will be influenced by both atmospheric N2O and biogenic N2O, it would be more useful to discuss these findings in terms of % saturation. Emissions of N2O from the system also depend on saturation dynamics (see classic description of N2O solubility in Weiss & Price (1980) Marine Chemistry).
It is outside the scope of this manuscript to quantify the release and background levels of N2O in soil water and drain water as we solely looks at changes in N2O between control plots and manipulated plots with controlled drainage in drain water N2O concentrations assuming that the background level of N2O at any time of year – both atmospheric N2O and biogenic N2O - is the same in soil water and groundwater in all 4 plots studied.

M. Section 2.2 says that groundwater ( ~7 piezometers per plot shown in Fig. 1) was sampled monthly for nutrient concentrations. However, the only groundwater data shown is the (unitless) annual nitrate value in Table 3. How variable were the concentrations over time? Did they differ between the control and treatment plots? How was groundwater data used to calculate N and P losses? What was the P concentration in groundwater?
Thanks for your comments. We agree that too sparse information is given about the nitrate concentrations in the upper groundwater, therefore a figure showing nitrate concentrations in the different piezometer pipes over time will be added to the supplementary information together with at map showing where the piezometers are located (by numbering the piezometers).
Groundwater data was only used to calculate nitrate losses, and the method applied to calculate nitrate loss is mentioned in the methods section 2.5 (line 9-10), however in line 9 it will be further specified that it is nitrate loss via groundwater we are referring to.

N.  More information is need on the spatial and temporal variability in other nutrient ´ parameters discussed (N2O, P, NH4+, SO4, DON, and PON). While some of this data is included in supplemental figures, the critical parameters should be included in the main manuscript in order to create a coherent and convincing story. This could be as simple as adding information on variability and sample numbers to Table 3.
Table 3 shows total loss in kilogram per hectare and the variability is given in the methods as the uncertainty related to flow measurements (±0.3%) and the detection limit of nutrient measurement (TN, nitrate and ammonium: 0.050 mgL$^{-1}$, TP and phosphate: 0.0005 mgL$^{-1}$) is stated.
An overview of sampling intensity will be added to the supplementary information (see answer B).

**Data interpretation**

O. Given the experimental design, this paper needs to be organised to more logically: explain how variables are, 1) different in treatment plots before and after induced conditions, and, 2) how treatment plots differed from control plots (i.e., where they the same prior to changed drainage conditions, as in, were the controls actually good controls?). The results and discussion are very disorganised, and the selective data displayed, make it hard to tease out the answer to either of these questions.
We hope that the changes we have suggested so far based on your comments will make it easier to answer the questions stated above.

P. The discussion around the NO3- isotope data is a bit hard to follow. First, it would ´ be useful to include a 95% CI for each slope described in Fig. 3 in order to more accurately judge if they overlap with the range expected for denitrification (1:1 – 2:1). As it seems that all of the data does plot roughly along a denitrification line, section 4.4 needs to be revised to discuss the data in terms of NO3- 'more impacted' v 'less impacted' by denitrification as values move up and down the denitrification line. It would then be useful to discuss what factors influenced these moves. As the authors note in the second paragraph of 4.4, denitrification is probably always occurring somewhere in an arable soil. It's therefore useful to keep in mind that the leached NO3- isotopes are a reflection of the degree to which denitrification is controlling the NO3- flux, and not direct measures of denitrification activity. This also means that it's a bit of an overstatement to say that higher NO3- isotopes show enhanced denitrification on a specific day. Instead, this higher value may indicate that reducing conditions dominated in the period prior to sampling (though, as this was only observed in one of the three plots, it also seems possible that this sample wasn't particularly representative of reality?). Overall the ~1:1 ratio of d18Ovd15N suggests that NO3- leached from the plots has undergone variable degrees of denitrification. So what controls these variations? Did isotope values increase in response to rainfall, season, temperature? And are these variations different between the control and treatment plots? I suggest checking out the paper recent advances in the interpretation of NO3- isotope data from, e.g., Hall et al. (2016) Oecologia and Wells et al. (2016) Water Resources Research when re-evaluating this data.
This section will be reevaluated. However the statement that denitrification occurred at a specific day is based on observations of nitrate, ammonium and sulphate concentrations and isotope analysis at this date, which all indicate denitrification, and not stable isotopes alone. In the article we suggest it could be due to the altered hydrology, increasing retention time and possibly creating anoxic conditions, but this we do not know this. We only see that drain flow is low around this time.
It must be kept in mind that the scope of the article was not to investigate the factors affecting denitrification, but to outline how controlled drainage affected water fluxes and pathways (tile drain and groundwater) within and from the plots studied, nitrate fluxes and pathways and any signs of changes in denitrification and nutrient swapping (impacts on N2O and phosphorus).

Q. The abstract and conclusion both mention 'pollution swapping', whereby decreases in NO3- leaching are countered by increases in N2O emissions. Here the drain N2O data is interested from the point of view of obtaining a more complete picture of N leaching

losses, but not conclusive evidence for/against pollution swapping. This is because soil surfaces are the primary source of N2O emissions (and thus the focus of concern in 'pollution swapping' follow drainage manipulation). Additionally, it is unclear if / how dissolved N2O was affected by controlled drainage, as in the first paragraph of section 4.4 it says that N2O-N was higher in the impacted plots, but then in the next paragraph it says that differences in N2O-N concentrations were not significant.

You raise a very valid point about pollution swapping, however we did primarily focus on pollution swapping with respect to losses via drain water, but we agree that nitrous oxide emission is the greatest from the soil surface, so we will revise this paragraph and include reference to study of measurements of N2O emissions. The loss of nitrous oxide from the surface was investigated by another research group and results from this study will be published this autumn (they found no difference between impacted and control plots).

Regarding how N2O-N was affected by controlled drainage the first sentence of section 4.4 (line 14-15) is about Y1, where nitrous oxide concentrations were significantly higher, while the next paragraph (line 25-26) is about Y2, where nitrous oxide where not significantly higher, but higher that at the control plots (Fig. S3 and S11, Table 2).

R.  The conclusions seem to say that the manuscript makes no contribution towards ´ understanding controlled drainage systems. A clearer case for why this manuscript should be published / read is needed.

This is another very important and highly useful point, and the conclusion will be rewritten emphasizing the novelty of this study regarding our findings and the need for further research on the topic.

Again, we appreciate all of your insightful and useful comments. We have tried to take into consideration all of your comments and will improve the manuscript accordingly. Again we are thankful to you for taking the time and energy to help us improve the paper.

---

## Author Comment (AC2) · 7 Oct 2016

Dear referee
We highly appreciate your constructive comments and suggestions that will surely improve our manuscript.

General comment
This paper focuses on the assessment of the impacts of controlled drainage on drain flow, groundwater levels and nutrient emission. The authors used a before after control-impact (BACI) sampling design in four adjacent drainage systems to test whether the controlled drainage had a significant impact on nutrient losses. They found that controlled drainage significantly affects the decrease in drain water flow and nitrate loss. The authors also combined the BACI experiment with a dual isotope approach (relation between _18O and _15N) to determine whether denitrification occurred in the impacted plots. The aim of the study is of interest for the readers of the journal and overall the paper is well written. Nonetheless, I suggest changes to the materials and methods section to add some important details, which are missing, and rephrase the conclusions to better highlight the novelty of the study.

We are delighted that you think the paper is of interest and well written, however we are now aware that we missed some important details both in the methods and results, and that the novelty of this study is not sufficiently emphasised in the conclusion. We will of course follow your advice and include more descriptions in the method section and rephrase the conclusion to better highlight the novelty of our study.

Specific comments
- The authors state that field management practices were similar during the three-year monitored period (lines 34-35, page 2), but at lines 22-24, page 5, they justify the lower nitrate concentration with the different agricultural management in the plots in 2011/2012. To avoid inconsistencies throughout the paper, I suggest to describe the field management practices carried out in the plots during the experiment (including the quantities of fertilizers) and clarify the possible effects of previous managements on the results obtained with the BACI experiment

We appreciate your suggestion and a management scheme including fertilizer amount and dates and harvest yield will be added (See the table next page)

| Plots | 2011/12 IP1, IP2, CP1 | CP2 | 2012/2013 (Y0) IP1, IP2, CP1 | CP2 | 2013/14 (Y1) IP1, IP2, CP1 | CP2 | 2014/15 (Y2) IP1, IP2, CP1 | CP2 |
|---|---|---|---|---|---|---|---|---|
| **Crop** | SB | WW | WW | WW | WW | WW | WW | WW |
| **Plowing** | 26 mar | 26 sep | 8 oct | 17 sep | 16 sep | 16 sep | | |
| **Sowing** | 27 mar | 27 sep | 9 oct | 18 sep | 17 sep | 17 sep | 18 sep | 18 sep |
| **Fertilizer application:** | | | | | | | | |
| Pig slurry | 10 may | 19 apr | 1 may | 1 may | 5 may | 5 may | 1 may | 1 may |
| -amount (ton) | 20 | 30 | 30 | 25 | 18 | 18 | 36 | 36 |
| Mineral, 1st | 27 mar | 16 mar | 15 mar | 8 apr | 26 mar | 26 mar | 20 mar | 20 mar |
| -amount (kg) | 103[a] | 156[b] | 125[a] | 125[a] | 200[c] | 200[c] | 150[d] | 125[d] |
| Mineral, 2st | | 20 apr | 20 apr | 9 may | 15 apr | 15 apr | 20 apr | 20 apr |
| -amount (kg) | | 172[b] | 194[b] | 194[b] | 215[c] | 165[c] | 100[b] | 100[b] |
| **Harvest** | 21 aug | 21 aug | 21 aug | 21 aug | 21 aug | 21 aug | 21 aug | 21 aug |
| -yield (hkg ha[-1]) | 81, 81, 82 | 105 | 98, 100, 97 | 86 | 98, 99, 99 | 99 | 80, 75, 81 | 75 |

SB=spring barley, WW=winter wheat, Fertilizer type: [a]NS 21-24, [b]NS 27-4, [c]NS 28-5, [d]NS 26-13.

- In lines 28-30, page 3, the authors refer to an intensive sampling campaign carried out in Y1 to assess whether the opening would lead to an increase in the release of nutrient enriched water. The results of this intensive campaign are not reported in the paper. Do these results support the findings and are they relevant for the paper? If they are not relevant for the paper, it is better to remove the sentence in the Materials and Methods to improve the clarity of the section.
The data will be added to an existing figure (Fig. 2) and the findings will be discussed more thoroughly in the paper.

- Sections 2.3 and 2.4 omit how the water samples were stored before the analyses, if they were filtered and analysed immediately after the collection. These details should be included in the two sections.
Thanks for pointing this out. The information will be added.

- Figure 1 reports that there are eight piezometers installed in each plot for groundwater level measurements (and water sampling), but in Fig. 2b there are only two series of dots. Do the dots represent an average groundwater level? If so, this information should be included in the caption and the authors should discuss the spatial and temporal variability of groundwater levels and nutrient concentrations and report which values (all the data collected?) they used for the BACI test (Table 2) and for the calculations of total losses of chemicals (Table 3). Furthermore, the description of the locations of piezometers in the plot, as reported in Table S3, is quite confusing. Is it

C2 possible to add letters/numbers in Fig. 1 or have another map in the supplementary material?

We are grateful that you emphasis this. This paper includes many different types of data, therefore it is very important that it is stated clearly which data is used in the respective analysis. We realize that we have not fully succeeded at this therefore it will be clarified in the revised paper.

At each plot 8-9 piezometers were located, but only the piezometers next to the regulation well was equipped with a pressure transducer measuring groundwater levels on a daily basis, which are the data shown in Fig. 2b. All of the other piezometers were measured every month. However, this will be emphasised in the paper and also a new Table with an overview of sampling frequencies and locations will be added (See Table below). A map with numbered piezometers will be added to supplementary information, and these numbers will be used in Table S3.

| | | |
|---|---|---|
| Plots with CD | IP1 and IP2 | |
| Plots without CD | CP1 and CP2 | |
| Management of regulation well at IP 1-2 | closed | opened |
| Y1
Y2 | 10-dec-13
17-nov-14 | 11-mar-14
09-mar-15 |
| Reference period | Y0 (21-nov-2012 to 21-apr-2013) | |
| Regulation level in
Period 2
Period 3 | 50 cm *
70 cm | |
| Number of piezometers pr. plot with pressure transducer | 1 | |
| Number of piezometers pr. plot without pressure transducer | 8 | |
| Frequency of water sampling in piezometers | 2-3 times a month | |
| Frequency of water sampling in the measuring well | Weekly | |
| Frequency of drain water flow measurement | Every 10th minute | |
| Frequency of ground water level measurements in piezometer with pressure transducer | Daily** | |
| Frequency of ground water level measurements in piezometer with continuous pressure transducer | 2-3 times a month | |

* until 28 January 2013 for CP1 hereafter 70 cm.
** Often lower frequency due to low inflow time of soil water, thus data from IP2 from all periods was unusable. Dysfunctional pressure transducer at CP1 in beginning of Y0 and at CP2 in Y3.

- The authors should explain why they replaced CP2 values with CP1 in the calculations for Table 3. Did the authors assume that the difference between the samples collected at the two control plots is not significant?

Thanks to your comments we realize that the footnote of Table 3 is poorly phrased. The data represented in the table is the results from CP2, however these results were omitted from the analysis (BACI and calculation of percentage loss) as winter wheat was grown at this field prior to the experiment (2010/11), while spring barley was grown at the other plots. The consequence was that N concentrations were much lower at CP2 compared to the other plots in 2012. Thus it was decided to omit the data from CP2 in Y0. This will be explained more clearly in the footnote.

In Section 4.4 (lines 16-18, page 8) the authors report the slope for the relation between _18O and _15N in Y0 and comment it. In order to improve the consistency and compare Y0 with Y1 and Y2, is it possible to add the data in Fig. 3?

Thanks for pointing this out. The data will be added to Fig.3.

- The Conclusions section reports briefly the main findings of the study, but the novelty is not very clear or is not highlighted as it should be. Therefore, I would recommend to rephrase the Conclusions.

Thanks for your suggestion. We will rephrase the conclusion and emphasis the novelty of the study.

Technical corrections

We appreciate your corrections and will incorporate all of them.

- Figure 2: Please report the origin of nitrate concentrations (drain water?).
- Figure 3: Measurement units are missing in the x and y axis. Please add them and zoom in to improve the readability of the figure.
- Table 1: Please add the measurement units and standard deviations whether average values are reported in the table.
- Table 2: Please report in the caption what 'b.d.l.' means.
- Table 3: Please add the measurement units and standard deviations whether average values are reported in the table.

Again, we appreciate all of your insightful and useful comments. We have tried to take into consideration all of your comments and will improve the manuscript accordingly. Again we are thankful to you for taking the time and energy to help us improve the paper.

---

## Author Comment (AC3) · 7 Oct 2016

Dear Mr. Joachim Rozemeijer
We highly appreciate your constructive comments and suggestions that will surely improve our manuscript.

General comments
This paper presents a solid study on the effects of controlled drainage on nutrient losses from an agricultural field. The paper is well written and well structured.
I have one major comment. The authors conclude that the water discharge and nitrate losses to surface water via the subsurface drainage system has considerably reduced after implementing controlled drainage. This raises the question where this water and nitrate goes instead. There was no influence on the harvest yield, so probably no extra evapotranspiration and crop uptake. Denitrification was also not markedly enhanced. The authors report that no overland flow was observed. The water and nitrate must have infiltrated to the upper groundwater. From there, the fate remains uncertain. The extra nitrate load may have polluted the deeper groundwater resources. In this case, pollution swapping did occur; less nitrate loss to surface water, more nitrate loss to groundwater. The other option is an enhanced shallow groundwater flow towards the surface water. In this case, there is no reduction of the nitrate load to surface water. The uncertainty about the fate of nitrate is described in the discussion (p7 L11-21) and in the conclusion (p9 L6-11). However, this crucial aspect is missing in the abstract, which only presents the positive effects of controlled drainage.
Thanks for stressing this aspect. We agree that the unknown fate of nitrate must be included in the abstract.

In the discussion, I would expect a more thorough discussion about the potential negative effects. Furthermore, an evaluation of the research methodology could be added to the discussion. How can the total effects of controlled drainage be quantified in future studies? The authors only suggest tracer additions and 3D modelling (p7L18-19). Could more intensive hydrological and chemical monitoring of different flow routes also add to this? A sentence evaluating the monitoring setup could also be added to the abstract and the conclusions.
Thanks for this comment; you raise a very important and highly useful point. We will revise the discussion with this in mind. With respect to how controlled drainage can be quantified in future studies, it will be difficult to give general recommendations as controlled drainage is used in very different ways and locations. However, we believe that knowledge of the location of the redox zone is of great importance with respect to assessing whether groundwater is reduced or not, which should also be included in the paper.

Minor comments: P1L12: 'For the first time': it's unclear what exactly was for the first time. A controlled drainage pilot in Denmark? A controlled drainage pilot on a field with winter crops? Controlled drainage as mitigation for nitrate losses? Etc.
Thanks for raising this point. This sentence will be rephrased.

P4L33-35: Could you add more information about the regulation level management of the controlled drainage system? This should be part of section 2.

We appreciate that you emphasis this. This has also been suggested by the other referees therefor we will add a scheme containing this information.

| Plots with CD | IP1 and IP2 | |
|---|---|---|
| Plots without CD | CP1 and CP2 | |
| Management of regulation well at IP 1-2 | closed | opened |
| Y1
Y2 | 10-dec-13
17-nov-14 | 11-mar-14
09-mar-15 |
| Reference period | Y0 (21-nov-2012 to 21-apr-2013) | |
| Regulation level in
Period 2
Period 3 | 50 cm *
70 cm | |
| Number of piezometers pr. plot with pressure transducer | 1 | |
| Number of piezometers pr. plot without pressure transducer | 8 | |
| Frequency of water sampling in piezometers | 2-3 times a month | |
| Frequency of water sampling in the measuring well | Weekly | |
| Frequency of drain water flow measurement | Every 10th minute | |
| Frequency of ground water level measurements in piezometer with pressure transducer | Daily** | |
| Frequency of ground water level measurements in piezometer with continuous pressure transducer | 2-3 times a month | |

\* until 28 January 2013 for CP1 hereafter 70 cm.
\*\* Often lower frequency due to low inflow time of soil water, thus data from IP2 from all periods was unusable. Dysfunctional pressure transducer at CP1 in beginning of Y0 and at CP2 in Y3.

P2L45: In addition to anoxic conditions, you also need organic matter or pyrite for denitrification.

Thanks for the suggestion. Other factors controlling denitrication will be addressed here as you suggest.

P2L13-17: These 'hypotheses' are formulated here as questions. Replace hypotheses with research questions?

We agree and will replace hypotheses with research questions.

P4L27-30: Higher and more fluctuation groundwater levels due to more evenly distributed precipitation events?
Thanks for noticing this. Fluctuation should have been fluctuating. What we meant to describe was that in Y2 the precipitation events were occurring more often during the whole period compared to Y0 and Y1, which led to higher groundwater and also more fluctuating groundwater levels.

P4L32:"The implementation: : :(Table 1 )" I don't understand how this follows from table 1.
The reference here should have been Table 2. This will be changed in the revised paper.

P4L35: 5 cm per day? Why per day?
It was stated this way to stress that the BACI effect is an average difference of all groundwater monitoring data (which ideally was per day, however due to practically problems we did not have data from all dates), however it is more confusing than helpful, so this statement will be rephrased.

Again, we appreciate all of your insightful and useful comments. We have tried to take into consideration all of your comments and will improve the manuscript accordingly. Again we are thankful to you for taking the time and energy to help us improve the paper.